# sPAP/PAAT Ratio as a New Index of Pulmonary Vascular Load: A Study in Normal Subjects and Ssc Patients with and without PH

Walter Serra [1,*] and Alfredo Chetta [2]

[1] Cardiology Division, Cardio-Thoracic and Vascular Department, University Hospital, University of Parma, 43121 Parma, Italy

[2] Department of Medicine and Surgery, University Hospital, University of Parma, 43121 Parma, Italy; chetta@unipr.it

* Correspondence: wserra@ao.pr.it

**Abstract:** In pulmonary hypertension (PH), the development of right ventricular (RV) dilatation and RV failure are signs of accelerated progression of the disease, resulting in an increased risk of cardiac death. Even the noninvasive assessment of systolic blood pressure in the pulmonary artery undertaken by echocardiography does not provide a measure of ventricle–pulmonary interaction. Some studies have shown the potential for echocardiography to indirectly evaluate pulmonary vascular resistance (PVR) and the acceleration time of pulmonary outflow (PAAT). We used systolic pulmonary artery pressure (sPAP) and pulmonary vascular resistance to develop an sPAP/PAAT ratio (strength/surface unit)/(time) for this study. From January 2017 to December 2018, 60 healthy subjects and 63 patients with systemic scleroderma (Ssc) (60 females, 3 males), 27 with PH and 36 without PH at two-dimensional echocardiographic/Doppler, were screened. In normal subjects, the mean sPAP/PAAT ratio was $0.26 \pm 0.063$, which indicated optimal pulmonary arterial ventricle coupling and biventricular function. The data derived from the analysis of the Ssc patients showed that those presenting pre-capillary PH at cardiac catheterization had an sPAP/PAAT ratio of $0.40 \pm 0.05$. There was a significant correlation between sPAP/PAAT with Walk Distance (WD) and PVR, but not with TAPSE. Interobserver variability was less than 5%. The sPAP/PAAT ratio is a new parameter that may indicate pulmonary vascular afterload and interaction, both in normal subjects and in patients with Ssc and PH.

**Keywords:** pulmonary circulation; pulmonary hypertension; echocardiography; scleroderma

## 1. Introduction

In pulmonary hypertension (PH), the development of right ventricular dilatation and failure (RVF) are signs of disease progression [1,2]. Both lung pressure values and the degree of heart failure are independently related to clinical worsening [3,4]. Although the noninvasive estimate of pulmonary artery systolic pressure (sPAP) by Doppler echocardiography has been well established [5–8] it does not measure right ventricular-arterial coupling (RV–AC) and pulmonary vascular resistance [9–13]. The acceleration time of pulmonary outflow (PAAT) should be an indirect measure of PVR [14,15]. The assessment of the size, geometry, and function of the right ventricle is influenced by the complex shape of the chamber, which requires assumptions through noninvasive techniques.

Despite the limitations, there has been a consensus to consider echocardiography the most immediate and simple approach to obtain reliable markers of the systolic function of the right ventricle in cardiopulmonary disorders. Among the variables derived from the echo used to evaluate the systolic function of the right ventricle, the tricuspid annular plane systolic excursion (TAPSE) is easily obtainable and weakly predictive of poor results in patients with PH [10,11]. A combined assessment of PAAT [16,17] and sPAP would provide

more physiological information. Consequently, we have hypothesized that the relationship between PAAT (time) and sPAP (developed pressure) could be proposed and used as an in vivo expression of right ventricular load and pulmonary pressure in relation to time.

## 2. Methods

### 2.1. Patients

From January 2017 to December 2018, 123 consecutive patients, 60 volunteers normal (N) and 63 with systemic scleroderma (Ssc), (60F, 3M) according to the Digital Engagement and for Early Control and Treatment (DETECT) Trial [18], underwent clinical, echocardiographic evaluation. Patients were monitored for worsening dyspnea and for cardiac mortality via hospital and outpatient medical chart review.

The inclusion criteria of scleroderma patients were history, signs, symptoms, and treatment in accordance with American College of Rheumatology (ACR)/European League Against Rheumatism (EULAR) current guidelines [18]. Other exclusion criteria were the inability to perform 6MWT and/or poor acoustic window. Enrolled patients were monitored in this prospective observational study. Imaging was performed using a Philips IE33 and a 5.2 MHz transducer (Philips Medical Systems, Andover, MA, USA). A two-dimensional Doppler examination was performed using views specifically designed to optimize the RV imaging [19]. To obtain the TAPSE, the apical four-chamber view was used, and an M-mode cursor was placed through the lateral tricuspid annulus in real-time. The TAPSE was measured as the peak excursion of the tricuspid annulus (millimeters) from the end of diastole to the end of systole. The PAAT was measured by PW-Doppler through the pulmonary valve jet from the short-axis view as the interval (msc) between the onset of ejection and the peak flow velocity (Figure 1).

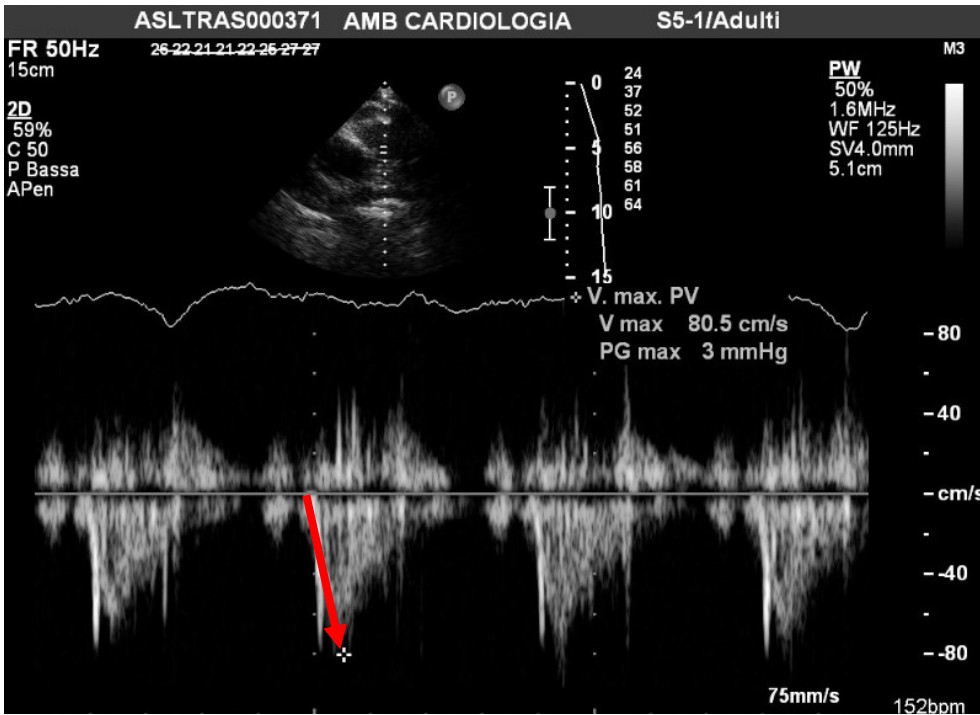

**Figure 1.** The PAAT was measured by PW-Doppler through the pulmonary valve jet from the short-axis view as the interval (msc) between the onset of ejection and the peak flow velocity.

The values were normalized for heart rate (PAAT/HR × 100). RV systolic pressure was determined from the tricuspid regurgitation (TR) jet velocity using the simplified Bernoulli equation and combining this value with an estimate of the correct atrial pressure by the diameter and collapsibility of the inferior vena cava, which was then added to

the calculated gradient for the sPAP. All measurements were performed by two blind senior researchers who read the recordings without knowledge of the clinical diagnosis. Subjects with significant valvular heart disease, hypertrophic or infiltrative cardiomyopathy, pericardial diseases, and unstable ischemic heart disease were excluded.

## 2.2. Right Heart Catheterization

The patients underwent right femoral heart catheterization using a 7F catheter thermodilution within two weeks of the echocardiographic exam to evaluate the systolic, mean, and capillary pulmonary pressure and calculate the transpulmonary gradient (TPG = PAPm − PCWPm), pulmonary vascular resistance (PVR:dynes/sec/cm$^{-5}$ = TPG/CO × 80) and cardiac output (CO) [1]. PH was defined as a mean PAP (PAPm) > 25 mm/Hg and a pulmonary capillary wedge pressure (PCWP) < 15 mm/Hg [2]. Although new values have been proposed and a new hemodynamic definition and updated clinical classification of pulmonary hypertension have emerged from the sixth World Symposium of PH [19,20], the authors chose to reject the new guidelines as they were reported after the completion of this study.

## 2.3. Statistical Analysis

The primary objective of the study was to assess the range of sPAP/PAAT ratios in healthy subjects and in patients with scleroderma and PH. The second objective was to correlate the sPAP/PAAT ratios and the hemodynamic parameters (sPAP, PVR) for the right heart catheterization. Data are expressed as mean values ± SD unless otherwise stated. Pearson's correlation coefficient and Bland–Altman analysis was used to compare PAP, PAAT, sPAP/PAAT ratio, and TAPSE with PAP, PVR, and NYHA class. Stepwise forward multiple regression analysis allowed the weighting of the independent effects of the potential determinants on an independent variable. The null hypothesis was rejected when the *p*-value was less than 0.05. Blinded to the clinical data to assess the interobserver variability, two sonographers measured the sPAP/PAAT ratio.

The area under the receiver-operating characteristic curve (AUC-ROC) was used to plot the true positive rate (i.e., sensitivity) as a function of the false-positive rate (specificity) for different cutoff scores of sPAP (mm/Hg) with respect to sPAP/PAAT ratio of 0.26, indicative of an optimal pulmonary arterial ventricle coupling and biventricular function, as a threshold value.

## 2.4. Ethics Statement

This study was conducted in accordance with good clinical practices and the current version of the revised Declaration of Helsinki. Written informed consent was obtained.

## 3. Results

Baseline demographic, clinical, and hemodynamic characteristics of patients with scleroderma and normal arterial pressure are summarized in Table 1, while baseline demographic, clinical, and hemodynamic characteristics of patients with scleroderma and PH are summarized in Table 2.

To determine the values and gender distribution, an echocardiogram was obtained from 60 normal volunteer subjects, mean age 55 ± 8 years.

In the population evaluated, the female sex was prevalent (>90%). There was a positive correlation between the NYHA class and sPAP/PAAT.

**Table 1.** Baseline demographic, clinical, and hemodynamic characteristics of patients with scleroderma without pulmonary hypertension.

| Total Cohort: |
| --- |
| (*N* = 36) |
| Age, years 55.6 ± 9 |
| Female sex 34 |
| NYHA I 26 pts; NYHA II 10 pts |
| **Echocardiographic Analysis:** |
| LV EF, %58 ± 6 |
| TR pressure gradient mmHg 26.5 |
| Mean PAAT 110 msc |
| TAPSE 23 mm |
| sPAP/PAAT 0.26 |

**Table 2.** Baseline demographic, clinical, and hemodynamic characteristics of patients with scleroderma and pulmonary hypertension.

| Total Cohort |
| --- |
| (*N* = 27) |
| Age, years 69.7 ± 8 |
| Female sex (26) |
| NYHA I 15 pts; NYHA II 9 pts; NYHA IV 3 pts |
| **Echocardiographic Analysis:** |
| LV EF, % 56 ± 6 |
| TR maximum pressure gradient, mmHg 46.5 ± 10 |
| Mean sPAP/PAAT 79.7 ± 7 |
| TAPSE 19.7 mm ± 10 |
| sPAP/PAAT 0.4 |
| **Invasive hemodynamic:** |
| Heart rate, bpm 85 ± 16 |
| RV systolic, mmHg 58 ± 20 |
| PA systolic, mmHg 60 ± 20 |
| Mean PAP, mmHg 37 ± 13 |
| Mean PCWP, mmHg 15 ± 2 |
| Cardiac output, L/min 4.7 ± 1.3 |
| Pulmonary vascular resistance: 5.63 W.U. |

Age was not significantly different between the two groups according to the Student's *t*-test. The sPAP/PAAT was weakly related to age, both in the healthy (r = 0.42, *p* = 0.014) and in the scleroderma patients (r = 0.41, *p* = 0.079). Interobserver variability was 5%. A total of 36 patients showed no PH and 27 with PH associated with Ssc, at a two-dimensional echocardiographic/Doppler evaluation. The sPAP/PAAT ratio showed high correlations with both the terms used for its calculation sPAP (R = 0.88, *p* = 0.001) and PAAT (R = −0.90, *p* = 0.001). In normal subjects, the mean sPAP/PAAT ratio was 0.26 ± 0.063, indicative of an optimal afterload with good pulmonary arterial ventricle coupling where pulmonary pressure, biventricular function, and NYHA class were normal. We then evaluated this parameter in patients with PH associated with Ssc. The data derived from the 20 Ssc patients with pre-capillary PH at cardiac catheterization had a sPAP/PAAT ratio of 0.40 ± 0.05 (Figure 2). There was a significant correlation among sPAP/PAAT, walk distance (WD), and PVR, but not with TAPSE. This could be attributed to sPAP/PAAT having a greater correlation with the afterload. Therefore, the ratio may be more useful than the load itself to determine the association to cardiac output, and arterial ventricle coupling (Figure 3a,b). A number of scleroderma patients showed combined pulmonary hypertension (Table 2). Taking all the patients into consideration, the AUC-ROC demonstrated an optimal threshold sPAP/PAAT of 0.4, which was predictive of PH.

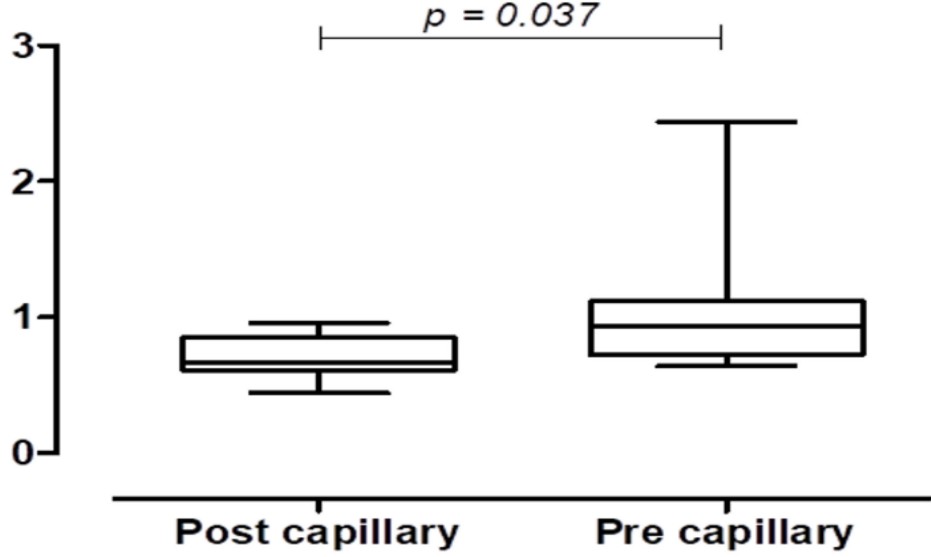

**Figure 2.** The initial data derived from the analysis of the 20 Ssc patients show that those presenting pre-capillary PH at cardiac catheterization had a sPAP/PAAT ratio >of 0.40 ± 0.05. Patients with post-capillary PH had a sPAP/PAAT >of 0.28 There was a significant correlation between sPAP/PAAT and WD, but not with PVR, perhaps due to this parameter having a greater correlation with cardiac output and arterial ventricle coupling.

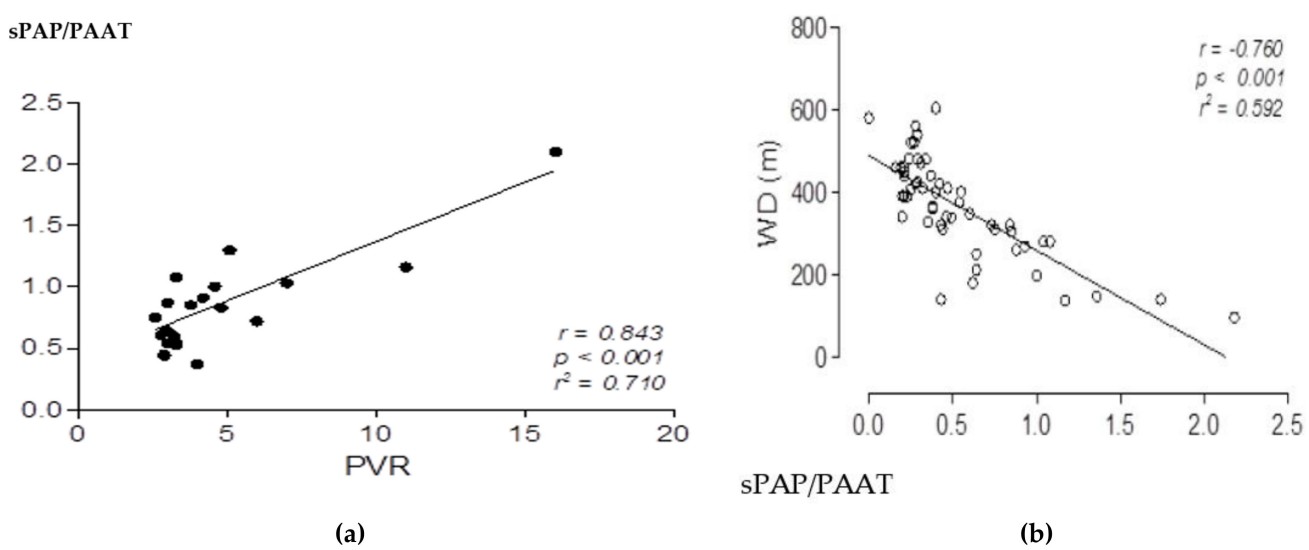

(a)                                                                                                      (b)

**Figure 3.** (**a**) Correlation between sPAP/PAAT and PVR. (**b**) Correlation between sPAP/PAAT and WD.

The sPAP cutoff point, which maximized sensitivity and specificity, was 35 mm/Hg (0.76 sensitivity and 1.00 specificity) (Figure 4).

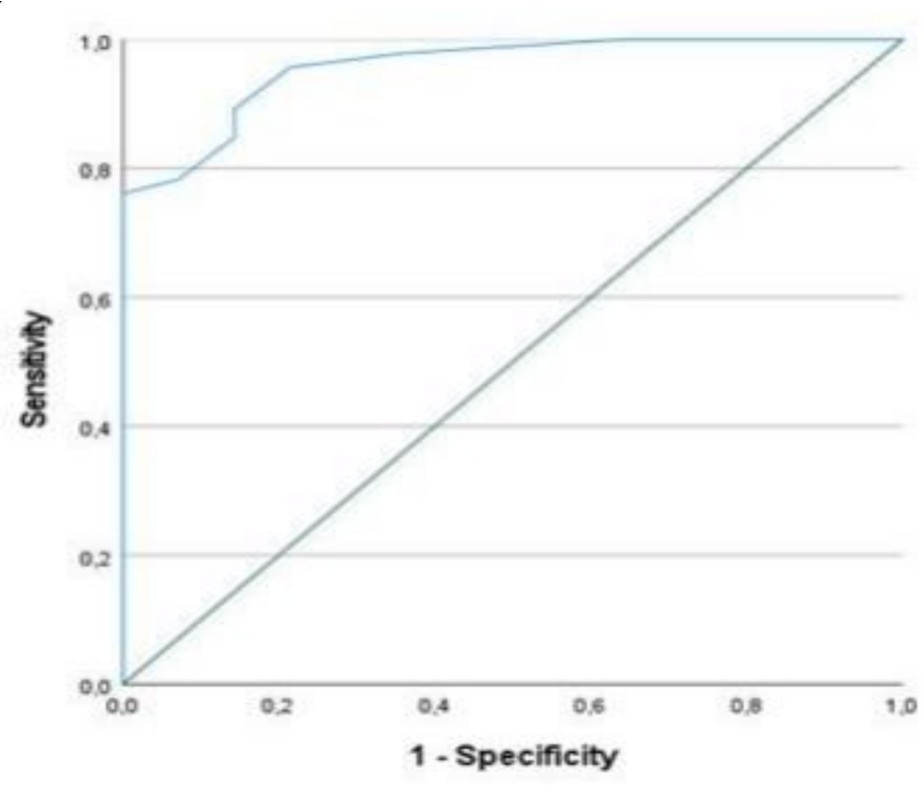

**Figure 4.** Area under the receiver-operating characteristic curve (AUC-ROC) method plotting the true-positive rate as a function of the false-positive rate for a cutoff point of sPAP with respect to a 0.26 ratio of sPAP/PAAT, as the threshold value, showed 0.957 AUC-ROC value (*p* = 0.0001).

## 4. Discussion

Our findings provided new insights into the RV functional evaluation in PH with both pathophysiological and clinical implications. The right ventricular arterial coupling (RV–AC) has been used to assess the association between contractility and afterload. It is calculated as the ventricular elastance (Ees) over arterial elastance (Ea) ratio (Ees/Ea). The invasive assessment, previously obtained from volume-pressure curves, led to practical noninvasive approaches such as echocardiographic and cardiac magnetic resonance imaging. The most robust RV–AC echo parameter was proposed by Guazzi et al. [21] using the TAPSE/sPAP ratio, where a numerator presented the RV performance, and a denominator presented the afterload parameter. An effective noninvasive correlation was determined to be the gold standard for RV–AC as the ratio of ventricular elastance to arterial elastance. Additional clinical and prognostic data have been reported over the last five years among varied pulmonary hypertension populations. Moreover, a study examining different echo parameters that may provide a noninvasive approach for RV–AC in Ssc was recently published [22].

Based on previous studies, we validated the echocardiographic sPAP/PAAT parameters versus the RHC-derived parameters. The sPAP/PAAT ratio represents the power required for the heart to push the blood into the pulmonary artery. Acceptable conditions are lower resistance that corresponds to a longer acceleration time. In this case, the heart provides low energy to the blood, so it works less. Doppler echocardiography-derived PAAT assesses the blood flow velocity characteristics in the RV outflow tract in response to changes in RV mechanical performance and pulmonary vascular load. Recent studies have demonstrated that PAAT provided a reliable estimate of invasive vascular resistance [23–25]. The novelty of our study was that the relationship of sPAP/PAAT was applied as a noninvasive index to characterize ventricular–vascular coupling showed all aspects of the RV, and could be useful for risk stratification and long-term monitoring.

In our study, there was no correlation between sPAP/PAAT and TAPSE; however, there was a significant correlation between the PVR and the WD, perhaps due to these parameters having a stronger association with the afterload, cardiac output, and arterial ventricle coupling. The ventricular work decreased by $0.19 \pm 0.063$ as the ratio decreased in normal subjects. In subjects with PH, the high ratio highlighted the increased effort of the right ventricle against high pulmonary vascular resistance. Both sPAP and PAAT are indirect afterload measures, so the ratio may be a better approach for afterload and pulmonary right ventricle interaction. This result indicated the possibility of measuring the efficiency of the right heart power before and after a pharmacological or surgical intervention, such as a pulmonary arterectomy. It has been well established that there is poor RV adaptation to overload in Ssc, which has also been linked to complex pathophysiology that could be identified by the sPAP/PAAT ratio.

To the best of our knowledge, this study was the first systematic assessment of sPAP/PAAT ratio in a cohort of healthy and scleroderma patients with and without PH by transthoracic echocardiography, date. Finally, this approach may help to define earlier stages of symptoms that reflect the early reduction of RV functional reserve.

*Limitations*

Our study was at a single center and relatively small. Moreover, echocardiographic examinations and RHC were not performed simultaneously, which may have affected the results. We found sPAP/PAAT values that correlated with pulmonary arterial hypertension, but these would need to be better evaluated further in studies that use cardiac catheterization in all patients. In our study, only 20 patients with PAH and scleroderma had a RHC performed. sPAP/PAAT did not correlate well with TAPSE, which may have been related to the susceptibility of TAPSE to the variability in image acquisition.

Finally, the determination of sPAP/PAAT was slightly limited by the availability of pulmonary Doppler signals.

## 5. Conclusions

The sPAP/PAAT ratio is a new parameter that may indicate pulmonary vascular load and pulmonary arterial ventricle interaction in both normal subjects and in patients with Ssc and with or without PH.

**Author Contributions:** Conceptualization, W.S.; methodology, W.S.; software, A.C.; validation, W.S. and A.C.; formal analysis, A.C.; investigation, W.S.; data curation, W.S.; writing—original draft preparation, W.S.; writing—review and editing, W.S.; supervision, A.C.; All authors have read and agreed to the published version of the manuscript.

**Funding:** This research received no external funding.

**Institutional Review Board Statement:** The study was conducted in accordance with the Declaration of Helsinki and approved by the Ethics Committee of University Hospital of Parma (protocol code 9/17 Right NETwork and date of approval 26 July 2017).

**Informed Consent Statement:** Written informed consent has been obtained from the patients to publish this paper.

**Data Availability Statement:** Data supporting reported results presented in the manuscript.

**Conflicts of Interest:** The authors declare no conflict of interest.

**Abbreviations:**

| | |
|---|---|
| PH | pulmonary hypertension |
| BMI | body mass index, defined as weight in kilograms divided by the square of height in meters |
| CAD | coronary artery disease |
| CI | confidence interval |
| DPG | diastolic pulmonary gradient |
| LV | left ventricle |
| PAAT | acceleration time of pulmonary outflow |
| mPAP | mean pulmonary artery pressure |
| PCWP | pulmonary capillary wedge pressure |
| PVR | pulmonary vascular resistance |
| RA | right atrium |
| RV | right ventricle |
| RV FAC | right ventricular fractional area change |
| sPAP | systolic pulmonary artery pressure |
| sPAP/PAAT ratio | systolic pulmonary artery pressure/acceleration time of pulmonary outflow ratio |
| Ssc | systemic sclerosis |
| TAPSE | tricuspid annular plane systolic excursion |
| TPG | transpulmonary gradient |
| TR | tricuspid regurgitation |
| RHC | right heart catheterization |
| WD | Walk distance |
| WU | Wood unit |

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
