# Peer review of "sPAP/PAAT Ratio as a New Index of Pulmonary Vascular Load: A Study in Normal Subjects and Ssc Patients with and without PH"

_pathophysiology, doi:10.3390/pathophysiology29010012_

Round 1

Reviewer 1 Report

The authors made some efforts to improve the manuscript. However, most of the concerns were not addressed at all. It seems that the manuscript and the revision were sloppily prepared.

The font sizing is still inconsistent. There is no consistency in using the decimal separator, it should be either the comma or the dot. I suggest a detailed copyediting of the manuscript by an expert in the field.

The authors should pay attention to the proper using of abbreviations. Abbreviations should be defined at first mention. In some instances, abbreviations are used without spelling them out (e.g., sPAP, 6MWT, RV, PVR, ACTPO etc.) or spelled out several times (e.g., RV-AC, WD). In the main text, figures and even within the same figure, the abbreviations are not consistent (e.g., sPAP, PAPs and PAPS, PAAP and ACTPO).

Introduction

Lines 36-37. The PAAT is an indirect measure of what parameter?

Line 45. TAPSE is predictive of poor outcome (not results) in patients.

Methods

It is not clear why patients with scleroderma were chosen for the study. It should be explained in the introduction.

Lines 76-82. The description is confusing. First of all, right ventricular systolic pressure is determined by adding estimated right atrial pressure to the tricuspid pressure gradient (line 79). Then right ventricular systolic pressure is assumed to be equal to systolic pulmonary artery pressure (line 81). Here, there is one more example of inconsistency in using abbreviations (sPAP vs. PAPs, lines 79 and 81). In addition, neither sPAP, nor PAPs was spelled out in the text.

Line 84. “Valvular heart disease” is the correct term

Lines 87-97. It is not mentioned that right heart catheterization was not performed on all patients. What were indications for conducting right heart catheterization?

Line 101. PVR was not spelled out.

Line 107. The null hypothesis is rejected when p value is less than 0.05.

Results

No data on 60 healthy volunteers are presented.

Line 128. Normal pulmonary artery pressure?

Table 1 and 2. No SD and units are reported for some parameters. All abbreviations used in tables should be defined in the table note.

Line 186. “Pulmonary”

The values for sPAP/PAAT ratio reported for healthy volunteers are different in abstract (line 21) and in results (line 205).

Figure 2. Axes labeling is not optimal and confusing. Measurement units are missing. The figure caption does not fully correspond to the figure. Moreover, it contradicts the statement in the main text and figure 3. Why data on only 20 initial patients presented here? What is the number of patients in each group? It seems, that sPAP/PAAT ratio is above 0.4 for both groups. What is the meaning of this figure?

Figure 3. Measurement units are missing.

Line 285. How can sPAP/PAAT values correlate with pulmonary arterial hypertension? They do correlate with some other parameters, such as PVR, WD.

Lines 286-287. These statements are controversial (compare with those on lines 202, 213-214).

Author Response

LETTER  point by point response to reviewers’ report

Thank you for the opportunity to resubmit the article after extensive review according to the instructions of the reviewers.

For the Editor

The authors made changes and upload as a new submission.

The authors responses to the critiques and explain why the changes were made .

  • The Title Page was changed.

  • English language editing of the text was made

  • We have deleted some words in the abstract .

  • The main text of the manuscript was changed and some sentences has been deleted

  • Table 1 and 2 written as text

For the Reviewers :

The authors thank the Reviewers for their valuable advice to provide a manuscript with greater potential for publication.

The following changes have been made :

Reviewer 1

Introduction

Lines 36-37. The PAAT is an indirect measure of Pulmonary Vascular Resistance (PVR)

Line 45. TAPSE is weakly predictive of poor outcome in patients with PH, because TAPSE is susceptible to the variability in image acquisition.

Methods

It is not clear why patients with scleroderma were chosen for the study:                                the patients were selected according to the DETECT Trial (18). Lines 76-82. The description is confusing. Because subjects had no significant RV outflow tract or pulmonic valve obstruction, RV systolic pressure was considered equal to sPAP :   This sentence has been deleted

Line 84. “Valvular heart disease” is the correct term

Lines 87-97. It is not mentioned that right heart catheterization was not performed on all patients. What were indications for conducting right heart catheterization?

Cardiac catheterization was performed according to the LG ESC / ERS of Pulmonary Hypertension Some patients did not consent to perform it.

Line 107. The null hypothesis is rejected when p value is less than 0.05.

Results

Line 128. Normal pulmonary artery pressure?   The sentence was thus changed:

 ….. without pulmonary hypertension

The values of the mean sPAP/PAAT ratio was 0.26±0,063, indicative of an optimal afterload. The value of 0.4 ± 0.05  was predictive of PH

Figure 2.  Why data on only 20 initial patients presented here?                                         Because only 20 patients have given consent to perform cardiac catheterization

 What is the meaning of this figure? The initial data derived from the analysis of the 20 Ssc patients shows that those presenting pre-capillary PH at cardiac catheterization have a sPAP/PAAT ratio > of 0.40 ± 0.05. Patients with post-capillary PH have a sPAP/PAAT > of 0,28

Figure 3. Measurement units are missing.: W.U. WOOD UNIT

Lines 286-287.  In our study only 20 p.ts   with  PAH and scleroderma performed a RHC

Thanks again for the detailed evaluation of the manuscript which therefore denotes some interest from the reviewers

Walter Serra MD,PhD

Reviewer 2 Report

sPAP/PAAT ratio: A new index of pulmonary vascular load.

This is a revision of a previously submitted manuscript. In this study, the authors sought to establish normative data based on a new measure sPAP/PVR in non-Ssc subjects and compare to normal subjects.

There are minor details that are incorrect but does not reflect a thorough check through the paper. I encourage the authors to read through the paper out loud, as odd sentence constructions distract from the science.

  1. Please clarify which pulmonary artery pressure does sPAP represent? I assume they authors mean systolic, but this needs to be clearly stated early in the paper. The authors so define it later in the methods.
  2. Methods, last sentence on page 2, the sentence “The PAAT was measured…to five beats.” What does that mean? Was it an average over 5 beats?
  3. Page 3, Section2.2. Please clarify that cath was performed within 2 weeks of…echo?
  4. How were normal subjects recruited? Criteria?
  5. Page 6, regarding age between the 2 groups. What 2 groups are the authors referring to? SsC +/- PH or SsC vs controls?
  6. Inconsistent numbers. There was mention of 27 SSc + PH patients but only 20 had cath data?
  7. Page 6, “Some of the scleroderma patients showed combined pulmonary hypertension.” What does this mean?
  8. Page 6, “In all patients, according to the ROC….” This can be written much more concisely. Most readers understand the point of ROC curves, so consider writing as “Taking all patients into consideration, the ROC curve demonstrates an optimal threshold sPAP/PAAT of 0.29.” Please clarify what this 0.29 indicates? Is it predictive of PH? Of poor outcomes?
  9. The fact that sPAP/PAAT does not correlate well with TAPSE may also be because TAPSE is susceptible to the variability in image acquisition. Maybe include in limitations.
  10. Page 9, “In subjects with PH, the high ratio tending to one…” what does “tending to one” mean in this context? Can we do without that phrase?
  11. Were there differences in sPAP/PAAT between SsC with and without PH?
  12. Figure 2 references 20 SSc patients who underwent cath. SSc is generally considered pre-capillary PH. Who then were in the post-capillary group?
  13. Did sPAP/PAAT correlate with WHO classification?

Grammatical edits:

  1. Abstract, the following sentence is poorly phrases: “From January 2017 to December 2018…patients…at two-dimensional echocardiographic/Doppler were screened.” Did the authors mean to say that patients who have had previous echo data were screened?
  2. Abstract, the word “was” can help the sentence read smoothly. “In normal subjects, we found that a sPAP/PAAT ration of 0.26 WAS indicative….”
  3. Introduction, second paragraph, the word “however” leads the reader to believe that the authors will make an argument against the previous statements regarding use of echo in evaluating PH. But there is no argument. Both supports the use of echo. Perhaps, a better way of wording this is “Despite the limitations, there is a consensus…..”
  4. Page 2, first sentence under Methods, “enrollment” is spelled incorrectly.
  5. Methods, page 3, last sentence of first paragraph, “All measurements were performed by two senior researchers who read the recordings exams blindly…” Please choose recordings or exams.
  6. “Data” is plural.
  7. Define abbreviations in Tables, example AcTPO. I don’t think in the paper, AcTPO was ever defined.
  8. Table 2, clarify that cath data represent mean values.
  9. Table 2, there is an unnecessary “)” when describing NYHA.
  10. Table 2, “Pulmonary” is spelled incorrectly.
  11. Figure 2: mislabeled sPAP as PAPs.

Author Response

LETTER  point by point response to reviewers’ report

Thank you for the opportunity to resubmit the article after extensive review according to the instructions of the reviewers.

For the Editor

The authors made changes and upload as a new submission.

The authors responses to the critiques and explain why the changes were made .

  • The Title Page was changed.

  • English language editing of the text was made

  • We have deleted some words in the abstract .

  • The main text of the manuscript was changed and some sentences has been deleted

  • Table 1 and 2 written as text

For the Reviewers :

The authors thank the Reviewers for their valuable advice to provide a manuscript with greater potential for publication.

The following changes have been made :

Reviewer 2

All required changes have been made are highlighted and reflected in the text

Title Page         sPAP/PAAT ratio: A new index of pulmonary vascular load.

  1. Please clarify which pulmonary artery pressure does sPAP represent? I assume they authors mean systolic, but this needs to be clearly stated early in the paper. The authors so define it later in the methods.
  2. Methods, last sentence on page 2, the sentence “The PAAT was measured…to five beats.” What does that mean? Was it an average over 5 beats?
  3. Page 3, Section2.2. Please clarify that cath was performed within 2 weeks of…echo?
  4. How were normal subjects recruited? Criteria?
  5. Page 6, regarding age between the 2 groups. What 2 groups are the authors referring to? SsC +/- PH or SsC vs controls?
  6. Inconsistent numbers. There was mention of 27 SSc + PH patients but only 20 had cath data?
  7. Page 6, “Some of the scleroderma patients showed combined pulmonary hypertension.” What does this mean?
  8. Page 6, “In all patients, according to the ROC….” This can be written much more concisely. Most readers understand the point of ROC curves, so consider writing as “Taking all patients into consideration, the ROC curve demonstrates an optimal threshold sPAP/PAAT of 0.29.” Please clarify what this 0.29 indicates? Is it predictive of PH? Of poor outcomes?
  9. The fact that sPAP/PAAT does not correlate well with TAPSE may also be because TAPSE is susceptible to the variability in image acquisition. Maybe include in limitations.
  10. Page 9, “In subjects with PH, the high ratio tending to one…” what does “tending to one” mean in this context? Can we do without that phrase?
  11. Were there differences in sPAP/PAAT between SsC with and without PH?
  12. Figure 2 references 20 SSc patients who underwent cath. SSc is generally considered pre-capillary PH. Who then were in the post-capillary group?
  13. Did sPAP/PAAT correlate with WHO classification?

Grammatical edits:

  1. Abstract, the following sentence is poorly phrases: “From January 2017 to December 2018…patients…at two-dimensional echocardiographic/Doppler were screened.” Did the authors mean to say that patients who have had previous echo data were screened?
  2. Abstract, the word “was” can help the sentence read smoothly. “In normal subjects, we found that a sPAP/PAAT ration of 0.26 WAS indicative….”
  3. Introduction, second paragraph, the word “however” leads the reader to believe that the authors will make an argument against the previous statements regarding use of echo in evaluating PH. But there is no argument. Both supports the use of echo. Perhaps, a better way of wording this is “Despite the limitations, there is a consensus…..”
  4. Page 2, first sentence under Methods, “enrollment” is spelled incorrectly.
  5. Methods, page 3, last sentence of first paragraph, “All measurements were performed by two senior researchers who read the recordings exams blindly…” Please choose recordings or exams.
  6. “Data” is plural.
  7. Define abbreviations in Tables, example AcTPO. I don’t think in the paper, AcTPO was ever defined.
  8. Table 2, clarify that cath data represent mean values.
  9. Table 2, there is an unnecessary “)” when describing NYHA.
  10. Table 2, “Pulmonary” is spelled incorrectly.
  11. Figure 2: mislabeled sPAP as PAPs.

Thanks again for the detailed evaluation of the manuscript which therefore denotes some interest from the reviewers

Walter Serra MD,PhD

Round 2

Reviewer 1 Report

No further comments